

# Close-range radar rainfall estimation and error analysis

Remco van de Beek[1,4], Hidde Leijnse[2], Pieter Hazenberg[3], and Remko Uijlenhoet[4]

[1]MeteoGroup, Wageningen, The Netherlands
[2]Royal Netherlands Meteorological Institute, De Bilt, The Netherlands
[3]Atmospheric Sciences Department, University of Arizona, Tucson, AZ, USA.
[4]Hydrology and Quantitative Water Management Group, Wageningen University, The Netherlands

*Correspondence to:* Remco van de Beek (remco.vandebeek@meteogroup.com)

**Abstract.** Quantitative precipitation estimation (QPE) using ground-based weather radar is affected by many sources of error. The most important of these are 1) radar calibration, 2) ground clutter, 3) wet radome attenuation, 4) rain induced attenuation, 5) vertical profile of reflectivity, 6) non-uniform beam filling, and 7) variations in rain drop size distribution (DSD). This study presents an attempt to separate and quantify these sources of error in flat terrain very close to the radar (1-2 km), where 4), 5),

and 6) only play a minor role. A 3-day rainfall event (25-27 August 2010) that produced more than 50 mm of precipitation in De Bilt, The Netherlands is analyzed using radar, rain gauge, and disdrometer data.

Without any correction it is found that the radar severely underestimates the total rain amount (by more than 50%). The calibration of the radar receiver is operationally monitored by analyzing the received power from the sun. This turns out to cause a 1 dB of underestimation. The operational clutter filter applied by KNMI is found to incorrectly identify precipitation

as clutter , especially at near-zero Doppler velocities. An alternative simple clutter removal scheme using a clear sky clutter map improves the rainfall estimation slightly. To investigate the effect of wet radome attenuation, stable returns from buildings close to the radar are analyzed. It is shown that this may have caused an underestimation of up to 4 dB. Finally, a disdrometer is used to derive event and intra-event specific $Z - R$ relations due to variations in the observed DSDs. Such variations may result in errors when applying the operational Marshall-Palmer $Z - R$ relation.

Correcting for all of these effects has a large positive impact on the radar derived precipitation estimates and yields a good match between radar QPE and gauge measurements, with a difference of 5 to 8%. This shows the potential of radar as a tool for rainfall estimation, especially at close ranges, but also underlines the importance of applying radar correction methods as individual errors can have a large detrimental impact on the QPE performance of the radar.

## 1 Introduction

Rainfall is known to be highly variable, both in time and space. Traditional measurements by single rain gauges or networks only provide accurate information of the rainfall at their locations. While interpolation of these data is possible the spatial information is often too sparse for accurate meteorological and hydrological applications (Berne et al., 2004; van de Beek et al., 2011a, b). Furthermore, rain gauges are often seen as 'ground truth', but these instruments also suffer from errors (Marsalek, 1981; Sevruk and Nešpor, 1998; Habib et al., 2001; Ciach, 2003).





Radar, on the other hand, does provide far better coverage in space and often also in time. However, a problem with radar systems is the larger number of error sources, which makes quantitative estimations based solely on radar difficult, unless these error sources are correctly addressed. Error sources that can be identified are radar calibration, clutter, wet radome attenuation, rain-induced attenuation, vertical profile of reflectivity (VPR), non-uniform beam filling (e.g. Battan, 1973; Fabry et al., 1992; Andrieu et al., 1997), and errors in derived surface rain rate from the measured reflectivity aloft due to uncertainties in the DSD (Uijlenhoet et al., 2003), the impact of wind drift and differences in instrumental characteristics (i.e. radar beam volume vs. point-based rain gauge). These variations in the error sources have been studied and described extensively in the past (e.g., Zawadzki, 1984; Hazenberg et al., 2011a, 2014).

Clutter results from the main beam or side-lobes (partially) reflecting off the terrain or atmospheric objects (e.g. buildings or trees, airplanes, insects, and birds). Close to the radar ground clutter from objects can lead to overestimation of rainfall reflectivities. Another source of clutter results from atmospheric conditions bending the emitted radar beam towards the surface (i.e. anomalous propagation ("anaprop")). This source of clutter can be highly variable in time, but its overall effect is generally limited. In the past many clutter correction schemes have been developed, which reduce the impact of clutter with varying degrees of success (e.g., Steiner and Smith, 2002; Holleman and Beekhuis, 2005; Berenguer et al., 2005).

Attenuation of the transmitted signal during a rainfall event can lead to strong underestimation of the rain rate. The amount of attenuation along the path of the transmitted signal is strongly dependent on the rain rate as well as on the transmitted wavelength. X-band radars are relatively inexpensive and easy to install, but suffer quite strongly from attenuation (e.g. van de Beek et al., 2010). Radars operating at longer wavelengths, like C-band and S-band, suffer less from attenuation. However, during intense precipitation events C-band radar rainfall retrievals also tend to underestimate precipitation rate (e.g. Delrieu et al., 1991; Bouilloud et al., 2009). Correction for rain-induced attenuation was first proposed by Hitschfeld and Bordan (1954). Since then other schemes have been developed that use a path-integrated attenuation constraint (e.g., Marzoug and Amayenc, 1994; Delrieu et al., 1997; Uijlenhoet and Berne, 2008). Another source of attenuation is caused by precipitation on the radar radome, resulting in a liquid film of water. This film attenuates the signal and its effect becomes more pronounced during stronger precipitation intensities. Wet radome attenuation is highly dependent on the wind direction and the state of the radome, as the attenuation depends on whether a film of water can form on the radome (Germann, 1999; Kurri and Huuskonen, 2008).

Vertical variations in precipitation as observed with radar give rise to the so-called vertical profile of reflectivity (VPR). The VPR has an important impact on the measurement characteristics of the radar. Especially for stratiform precipitation, the melting of snow flakes and ice crystals aloft results in relatively large droplets. Within this melting layer region, the returned radar signal intensifies (bright band), leading to an overestimation of the precipitation intensity (e.g., Andrieu et al., 1995; Vignal et al., 2000; Delrieu et al., 2009; Hazenberg et al., 2013). However, close to the surface, the role of the VPR tends to be limited.

Non-uniform beam filling can also cause significant errors. This effect of course depends on the size of the radar measurement volume and the spatial heterogeneity of the rainfall. Because the relation between radar reflectivity and rainfall intensity is non-linear and not unique (depending on the DSD), spatial rainfall variability within the radar measurement volume can



cause errors (Fabry et al., 1992; Berne and Uijlenhoet, 2005; Sassi et al., 2014). The DSD also directly influences the relation between the radar reflectivity and specific attenuation. In case rain-induced attenuation is not corrected for, the radar product is prone to result in erroneous rainfall estimates (Gosset and Zawadzki, 2001).

The conversion from measured reflectivity values to rain rates at ground level can be quite challenging as rain is highly variable in terms of its DSD (e.g., Uijlenhoet et al., 2003; Yuter et al., 2006). In general, the reflectivity value ($Z$) is converted into a rainfall rate ($R$) using a power-law relation:

$$Z = aR^b \tag{1}$$

To date, the Marshall-Palmer (M-P) equation (Marshall et al., 1955) with $Z = 200R^{1.6}$ is the most commonly used $Z-R$ relationship and is generally assumed to be representative for stratiform precipitation. It should be noted that other $Z - R$ relations have been derived as well, more suitable during different types of precipitation and for other locations (e.g., Battan, 1973; Fulton et al., 1998; Uijlenhoet, 2001; Uijlenhoet and Berne, 2008). Estimates of the DSD can be obtained by surface disdrometers, from which both $Z$ and $R$ can be inferred. Based on these estimates, it then becomes possible to infer the actual $Z - R$ relationship for the event of study at the location of the instrument (e.g., Löffler-Mang and Joss, 2000; Berne and Uijlenhoet, 2005; Hazenberg et al., 2011b). However, the benefit of applying disdrometer observations for weather radar rainfall correction application is still uncertain. As their point-based character might not be representative for the larger scale precipitation system aloft. Hazenberg et al. (2014) showed that using a disdrometer to determine the actual $Z - R$ relation did lead to improved results for convective precipitation. At the same time, making use of disdrometer observations for The Netherlands was shown to lead to improved precipitation estimates for widespread stratiform precipitation (Hazenberg et al., 2015)

This paper studies the possibilities of quantitative precipitation estimation (QPE) at close ranges (1-2 km) for a C-band weather radar operated by the Royal Netherlands Meteorological Institute (KNMI) in the center of the Netherlands. At these distances the effects of VPR, rain-induced attenuation, and non-uniform beam filling are limited. Their impact was therefore ignored in this work. Section 2 describes the instruments and the data used in this study, which are the same as used by Hazenberg et al. (2014). Section 3 describes the rain event that is analyzed. in Section 4 the reflectivity correction methods and their effects are discussed together with a verification. Finally, Section 5 contains conclusions and recommendations.

## 2 Instruments and data

The precipitation event analyzed in this paper was observed by the radar during the late afternoon on August 25 2010 and lasted for about 2 days and is well-known for the large amount of precipitation that fell, especially in the east of The Netherlands at Hupsel (Brauer et al., 2011; Hazenberg et al., 2014). A number of instruments, located at KNMI in de Bilt, the Netherlands, are used in this paper. These are a rain gauge, two optical disdrometers and an operational C-band Doppler weather radar. The instruments are located on a field south of the radar at KNMI. The instrument locations as well as the radar distance bin that has been used for the comparison are shown in Fig. 1.



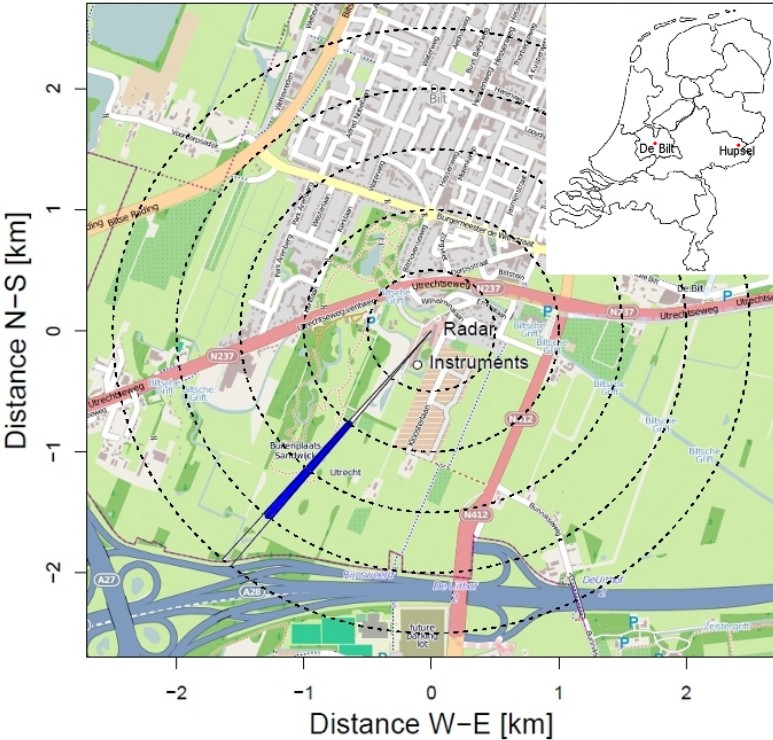

**Figure 1.** Locations of the rain gauge, disdrometers and radar at de Bilt with 0.5-km range rings around the radar (dashed lines). The blue section is the 1 to 2 km radar bin that is used in this study. The inset in the upper right corner shows the locations of radar and instruments at de Bilt and of Hupsel, which is the location of maximum rainfall measured during the event studied. Data by OpenStreetMap.org contributors under CC BY-SA 2.0 license.

The employed rain gauge is an automatic gauge with a surface area of $400 \pm 5$ cm$^2$ installed in a pit (Wauben, 2004, 2006). The height of a float in the reservoir of the gauge is measured every 12 seconds with a resolution of 0.001 mm. The gauge can report the precipitation intensity in steps of 0.006 mm h$^{-1}$. The rain is accumulated and stored at 10 minute intervals, using guidelines set by Sevruk and Zahlavova (1994) and WMO (1996).

The disdrometers are an OTT Parsivel and a Thies Laser Precipitation Monitor (LPM). They both measure the size and velocity of droplets by the extinction caused by droplets passing through a sheet of light with a surface area of around 50 cm$^2$. The Parsivel measures particles from 0.2 to 25 mm diameter with velocities between 0.2 and 20 m s$^{-1}$. The LPM is able to measure particles between 0.16 and 8 mm in diameter and velocities between 0.2 and 20 mm s$^{-1}$. For both instruments the beam between transmitter and receiver has been oriented perpendicular to the prevailing southwesterly wind direction in

the Netherlands. The data from the disdrometers are logged every minute (de Haij and Wauben, 2010). In the current work, observations obtained by the LPM disdrometer were only used as an additional source of precipitation information. Since the



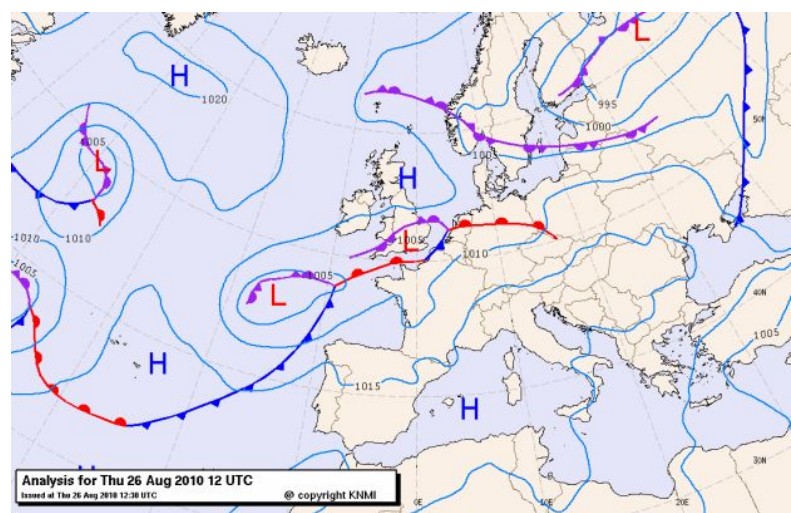

**Figure 2.** Synoptic situation for 26 August 2010 12 UTC.

**Table 1.** Daily precipitation sum and duration on 25–27 August.

|  | De Bilt | | Hupsel | |
| --- | --- | --- | --- | --- |
|  | sum [mm] | duration [h] | sum [mm] | duration [h] |
| 25 Aug | 6.4 | 5.7 | 1.6 | 3.0 |
| 26 Aug | 50.6 | 18.3 | 142.3 | 19.5 |
| 27 Aug | 3.0 | 5.5 | 14.6 | 4.5 |
| total | 60.0 | 29.5 | 158.5 | 27.0 |

observed $Z$ and $R$ measurements were similar between both instruments (not shown here), it was decided only to make use of the Parsivel disdrometer observations to study the impact of DSD variations.

The radar operated by KNMI is a Doppler C-band radar from SELEX-SI (Meteor AC360). It is located at 52.108N, 5.178E on top of a tower at 44 m above sea level. It operates at 5.6 GHz (wavelength of 5.3 cm). The radar performs a full 14-elevation volume scan every 5 minutes. The resolution is $1°$ in azimuth and 1 km in range. For details about the radar and the scan schedule, see Beekhuis and Holleman (2008). For this study we use the first distance bin between 1 and 2 km from the radar at an azimuth of $230°$ (see Fig. 1) of the $0.8°$ elevation scan.





## 3 Description of the rain event

Between 25 and 27 August 2010 a narrow band of low pressure passed over the Netherlands from the direction of the English Channel towards southern Denmark between high pressure zones over southern Europe and Scotland. During 26 August the triple point remained near the southern coast of the Netherlands for most of the day with the warm front moving very slowly

northward. This caused large temperature differences in the Netherlands between the north, with cold air, and in the south, with warmer air behind the warm front. During the afternoon of the 26th the low pressure zone began moving eastwards leading to quieter weather (see Fig. 2).

During the passage of these low pressure areas a mesoscale convective system containing large fields of alternately stratiform and convective precipitation passed over the Netherlands. This lead to both large precipitation amounts and long durations for

most of the Netherlands. Table 1 illustrates the amounts and durations for de Bilt, where the radar and instruments are located, and for Hupsel, located in the east of the Netherlands. At Hupsel an extremely large amount of precipitation of nearly 160 mm within 24 hours was measured (return period >1000 y) for this event (see Brauer et al., 2011; Hazenberg et al., 2014). At the location of the radar in De Bilt, which is the focus of this study, the total precipitation accumulation was less, although still considerable, with 50.6 mm over a period of 18.3 hours of continuous rain (return period 5-10 y; see Overeem et al., 2008,

2009).

The time series of precipitation is shown in the top panel of Fig. 3 and in Fig. 4. There is no precipitation until the late afternoon on the 25th. A long period of rain, with low to moderate rain rates, were observed at De Bilt (episode 1 in Fig. 3). The highest intensity cores were mostly observed just south of the radar and therefore not observed by the surface instruments used here. After a short dry period more precipitation passes over the radar with variable intensities. This period has been

subdivided into two phases. A first phase with moderate intensities of around 5 mm h$^{-1}$ (episode 2 in Fig. 3), and a second one containing heavier rainfall rates up to 25 mm h$^{-1}$ (episode 3 in Fig. 3). This period also gave rise to the largest number of raindrops measured by the Parsivel disdrometer (see Fig. 4). The large peak in episode 4 was the edge of an active squall line that began to form south of the radar and was advected eastwards, which caused large precipitation sums near Hupsel (Brauer et al., 2011). For episodes 5 to 8 rain intensities decreased within the trailing stratiform part of the squall line, resulting

in sporadic rainfall observed close to the radar. The total accumulations are shown in the bottom panel of Fig. 3. The two disdrometers and gauge are closely related, but the radar clearly underestimates rainfall accumulations.

## 4 Methodology and results

As explained in the introduction, various error sources affect rainfall measurements by weather radar. Since this work focuses on the performance of the weather radar at close ranges, it was decided not to focus on the impact of rain-induced attenuation,

VPR, calibration, ground clutter, and wet radome attenuation, as at close ranges these are expected to be negligible. Therefore, the current section specifically focuses on the effects of correcting for calibration, ground clutter and wet-radome attenuation. Furthermore, this section also presents the impact of accounting for DSD variations as inferred from disdrometer observations.



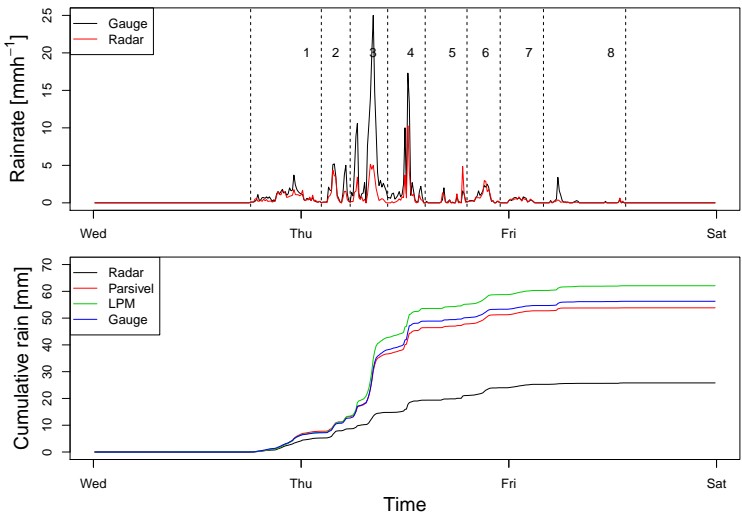

**Figure 3.** Top panel: time series of the rain event with the rain rate from the rain gauge in black and in red the rain rate derived from the radar reflectivity using the Marshall-Palmer $Z - R$ relation (M-P). The vertical dashed lines divide the event into 8 different episodes. Bottom panel: Cumulative sum of rainfall from the four instruments before any correction of the radar and using the M-P relation

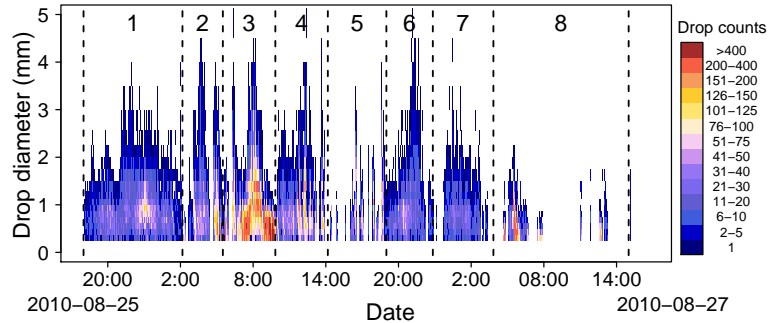

**Figure 4.** DSD during the rain event. The black dashed lines illustrate the identified rain episodes of the event.

### 4.1 Calibration

As explained in the introduction, the absolute radar calibration can have an impact on the QPE performance of weather radar (Ulbrich and Lee, 1999; Serrar et al., 2000). In the current work, we make use of the sensitivity of the receiver and the alignment of the radar to get accurate information on the possible calibration issues for the current event. The emitted signal from the sun is easily detectable by the radar as it is constant over all range bins. This signal can then be used to monitor the absolute calibration of the radar. This method is used operationally by KNMI (Holleman et al., 2010). These analyses showed that





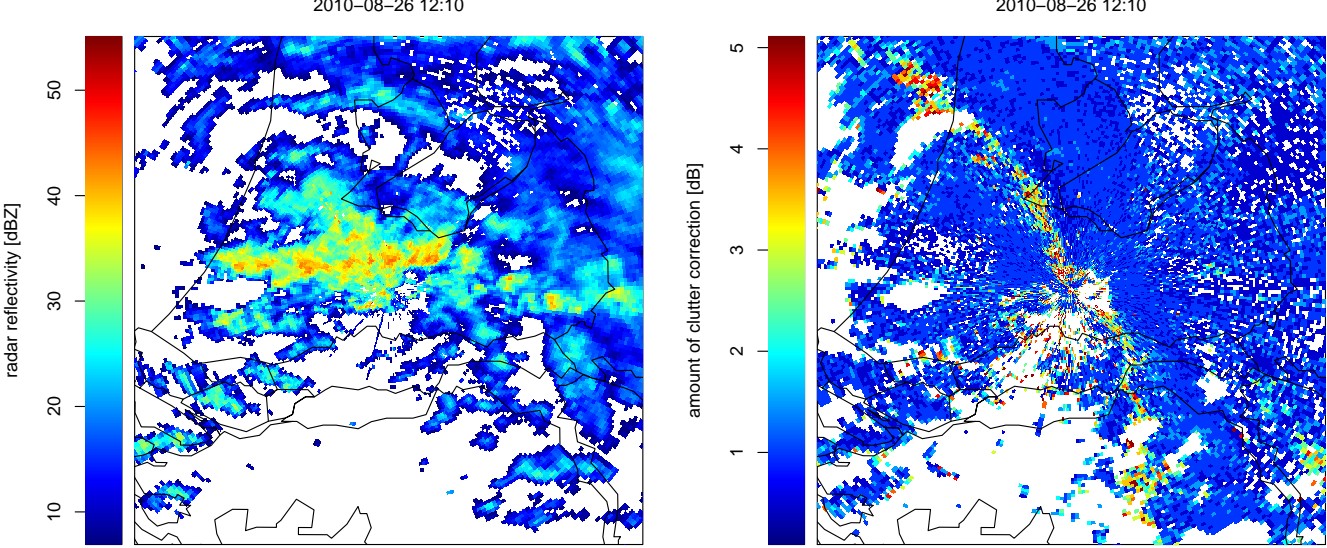

**Figure 5.** Radar reflectivity (left panel) and amount of clutter corrected by the Doppler notch filter (right panel) for the most intense rainfall peak of episode 4 on 26 August 2010 at 12:10 UTC. Images shown are for the 0.8° elevation scan.

the receiver calibration was off by 1 dB, resulting in an underestimation by the radar. The transmitter calibration is regularly checked and is therefore assumed to be correct. To account for this error source, a value of 1 dB was added to the observed radar reflectivity values.

### 4.2 Clutter correction

The operational ground clutter correction algorithm uses a time-domain Doppler notch filter. A drawback of this automatic procedure is that it incorrectly identifies some precipitation as clutter (e.g. Hubbert et al., 2009), leading to an underestimation of rainfall intensities as measured by the radar. An example is clearly shown in Fig. 5, where images of both the radar reflectivity factor and the amount of clutter correction are shown. The zero-isoDop, the region where the speed is perpendicular to the radar and therefore zero, is clearly visible in the right-hand panel of this image, and the amount of filtering in such areas can be as

high as 3-4 dB. For other areas, the amount of incorrect identification of precipitation as ground clutter is limited, although its effect can still be significant, on the order of 1-2 dB (i.e. a factor of 1.15-1.33 in terms of rainfall intensity given the Marshall-Palmer $Z - R$ relation).

  As an alternative procedure to correct for the impact of ground clutter, usage was made of a dry weather clutter map, consisting of the average dry weather reflectivity value. To correct for the impact of ground clutter during the precipitation event, this map

is subtracted from the observed reflectivity values. A main underlying assumption of making use of a static dry weather clutter map is that the clutter reflectivity does not change during rain (e.g. because objects become wet). This method (also called a clutter map) will not remove all clutter, however, it identifies less precipitation as clutter compared to a Doppler filter.





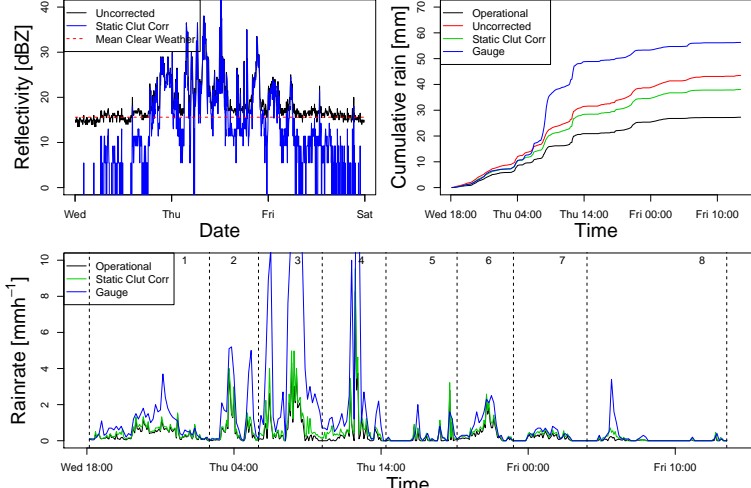

**Figure 6.** Top left panel: Reflectivity of the studied range bin between 1 and 2 km from the radar of the uncorrected reflectivity (black) and the static clutter corrected reflectivity (blue). Here the red dashed line is the average reflectivity when there is no rain. Top right panel: The cumulative rain sums for the rain gauge (blue) and the Doppler clutter corrected (black), together with the uncorrected (red) and the static clutter corrected reflectivity (green) using the Marshall-Palmer $Z - R$ relation with 1 dB added to compensate for calibration errors. Bottom panel: Time series of the rain gauge, Doppler clutter corrected and the static clutter corrected rain rates.

Fig. 6 illustrates the effect of the operational clutter removal scheme and using the static dry weather clutter map. In the top-left panel the raw uncorrected reflectivity values are shown in black. It can be observed that the average background reflectivity values during clear sky situations are around 15.6 dBZ (dashed red line). Subtraction of the mean value of Z (i.e. not dBZ) from the uncorrected reflectivity results in the simple static clutter removal (blue line). This has the greatest impact for

low reflectivities, with no or very little rain.

In the top-right panel of Fig. 6 the cumulative rainfall sums are shown for both rain gauge and radar rainfall (using the M-P relation) data. Radar accumulations are shown without clutter correction, and after applying either a operational Doppler scheme or the static dry-weather clutter correction method. Note that these results are obtained after applying a 1 dB calibration

correction. Results show that the uncorrected radar reflectivities produce the largest rainfall accumulations, as results are overestimated due to the identification of ground clutter as precipitation. Of the two clutter correction schemes, applying the the operational Doppler scheme results in the largest reduction of precipitation, whereas the static scheme is more conservative. As explained before, it is anticipated that the operational Doppler scheme incorrectly identifies some precipitation as ground clutter and as such results in the lowest precipitation accumulations. Therefore, in the remainder of the paper usage is made of

the static clutter correction scheme.



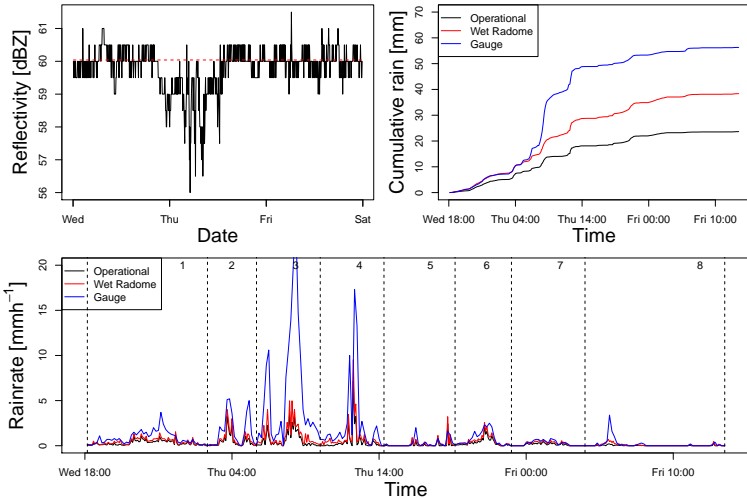

**Figure 7.** Top-left panel: Reflectivity of a strongly reflective clutter pixel near the radar. Here the red dashed line is the average reflectivity when there is no rain. Top right panel: The cumulative rain for the rain gauge (blue), the Doppler clutter corrected (black) and the wet-radome attenuation corrected rain rate using the Marshall-Palmer $Z - R$ relation using the static clutter and calibration corrected data (red). Bottom panel: Time series of the rain gauge, KNMI (Doppler filter) clutter corrected and the wet-radome attenuation corrected rain rates.

In the bottom panel of Fig. 6 the time series of the Doppler and static clutter corrected radar derived rain rates are shown together with those of the rain gauge. As expected, the static clutter corrected time series shows higher rainfall intensity peaks than those of the Doppler corrected time series and is generally closer to the rain gauge measurements. The small dip that is present in the peak of the Doppler corrected rain rate with very heavy rain in episode 4 of Fig. 6 disappears in the static clutter

corrected time series. This is a good illustration of the Doppler clutter removal scheme being too sensitive at times. There are a few exceptions to the underestimation by the radar, most notably the two highest peaks in episode 5, where the radar actually overestimates the rain rate compared to the rain gauge. A possible cause might be that the studied range bin lies further south than the other instruments, located at the measurement field of KNMI, and most of the strongest precipitation passed just south of the radar, especially during the formation of the squall line at the end of the rain event.

**4.3 Wet radome attenuation**

Since for the current event, precipitation with considerable intensities was observed at the location of the radar for a large period of time, it is highly likely that the resulting formation of a thin layer of water on top of the radome caused significant attenuation of the signal. The effect of the wet radome needs to be corrected and is achieved by using a strong clutter pixel observed close to the radar, caused by a tall building. Due to its close proximity to the radar (only 3 km away), it is assumed

that the impact of the rain-induced attenuation is negligible. We relate a decrease in the measured reflectivity value of this





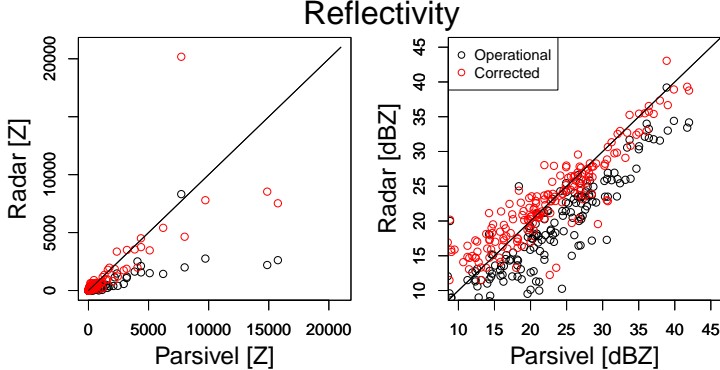

**Figure 8.** Left panel: Reflectivity ($Z$) measured by the radar and derived from the Parsivel where the black circles are the operationally corrected values and the red circles the fully corrected data described in this paper. Right panel: same as left panel but on a logarithmic scale.

static clutter pixel during a precipitation event to the amount of attenuation caused by the wetting of the radome. While the wetting of the clutter object and precipitation at the clutter location may also influence the measured reflectivity, these factors are assumed to be much smaller than the effect of the wetting of the radome.

Fig. 7 presents the impact of wet radome attenuation on the measurement capabilities of the radar. The top-left panel shows the measured reflectivity from the clutter pixel at a range between 3 and 4 km from the radar. The dashed red line presents the average reflectivity during dry periods. The reflectivity can be seen to fluctuate by about 0.5 dB around this mean value, however a larger drop in measured reflectivity values can be observed at the onset of the event in the late afternoon on 25 August. The difference between the average dry and observed reflectivity values is assumed to represent the impact of wet radome attenuation, which reaches its greatest value during the peak of very heavy rainfall. After having corrected for calibration error and clutter, the impact of wet-radome attenuation correction is shown in the top-right panel of Fig. 7. As expected the correction of the attenuated radar reflectivity results in a larger estimated rain rate, closer to that of the rain gauge.

## 4.4 $Z - R$ relations

The corrections applied so far all had a positive impact on the radar QPE performance. As a last aspect, the current section focuses on the impact of DSD variations. Figure 8 compares the reflectivity measurements of the radar to those inferred from the disdrometer. The corrections clearly have a positive impact, especially for high values of reflectivity (left panel of Fig. 8). If the M-P $Z - R$ relation is used, the accumulated rainfall increases from 25.8 mm for the uncorrected data to 47.1 mm after applying the corrections for calibration error, ground clutter and wet-radome attenuation (see Fig. 7). While this is still below the accumulated rain sum of 56.3 mm for the rain gauge, the net effect is considerable. Since the current precipitation event was highly variable in space and time, the applied M-P relationship is expected not to be suitable as it is representative for stratiform





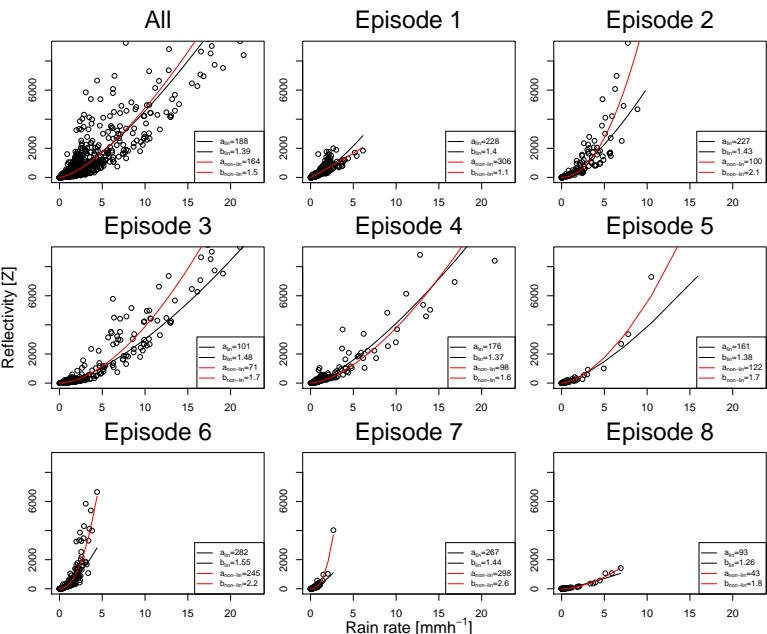

**Figure 9.** $Z - R$ relations derived from the one-minute data of the Parsivel for the different rainfall episodes distinguished in Fig. 3 using linear regression on the logarithmic values (black curves) and non-linear regression (red curves).

precipitation conditions. Therefore, further improvements in the quality of the rainfall estimates by radar can presumably be obtained using the $Z - R$ relationship inferred from the disdrometer measurements.

### 4.4.1 $Z - R$ relation derivation

Both the radar reflectivity $Z$ [mm$^6$ m$^{-3}$] and the precipitation intensity $R$ [mm h$^{-1}$] can be expressed as integral variables of the raindrop size distribution $N(D)$ [mm$^{-1}$ m$^{-3}$], where

$$Z = \frac{10^6 \lambda^4}{\pi^5 |K|^2} \int_0^\infty \sigma_B(D) N(D) \, dD, \tag{2}$$

and

$$R = 6\pi \times 10^{-4} \int_0^\infty D^3 v(D) N(D) \, dD. \tag{3}$$

Here, $\lambda$ [cm] is the wavelength at which the radar operates, $\sigma_B$ [cm$^2$] is the backscattering cross-section (proportional to $D^6$ in the Rayleigh regime; $\lambda \gg D$), and $v$ [$m\ s^{-1}$] is the terminal raindrop fall velocity. Hence both $Z$ and $R$ are functions of the DSD.





The relation between radar reflectivity and rainfall intensity can be expressed as a power-law function Battan (1973):

$$Z = aR^b \qquad (4)$$

The observations obtained by the Parsivel disdrometer are analyzed in more detail here, as from the measurement taken by this instrument joint estimates of $Z$ and $R$ are obtained. These observations enable one to study the impact of event and intra-event based variations of the $Z-R$ relation different from the M-P relationship.

In Fig. 9 the $Z - R$ relationships which are obtained from the one-minute disdrometer data for each of the identified episodes are presented. Multiple methods have been presented to derive the $Z-R$ relationship (Morin et al., 2003; Chapon et al., 2008; Hazenberg et al., 2011b). The simplest approach is to apply a least squares linear regression procedure on the logarithms of $Z$ and $R$. However, this approach tends to give more weight to smaller rainfall intensities. Therefore as a second approach, in the current work also a non-linear least squares fitting procedure was applied. From Fig. 9 it can be observed that the applied fitting technique has a large impact on the estimated values of the prefactor $a$ and and exponent $b$. The $Z - R$ relation varies greatly between episodes. For episode 1 a clear split in the observed $Z-R$ values is visible, suggesting that this episode can be better represented by two separate $Z - R$ relations. However, on the basis of an in-depth analysis using both the radar pseudoCAPPI images (not shown here) as well as the DSD data (see Fig. 4), it was impossible to accurately distinguish between these periods. Therefore, it was decided to treat this as a single episode. For the other episodes, such a clear distinction cannot be observed, although some episodes show more scatter than others.

In the remainder of this work it was decided to apply the estimates of $a$ and $b$ (see Eq. 4) obtained by the non-linear least squares approach, as higher values obtain a larger weight by this procedure.

The non-linear power-law fits, together with the Marshall-Palmer $Z - R$ relation, are shown in the left panel of Fig. 10. For the current rainfall event, the optimal $Z - R$ relationship varies considerably between the different episodes. As expected, the Marshall-Palmer relation is not representative for any of the eight episodes. Therefore, applying this relation results in an overall underestimation of the actual rain rate by the weather radar. This especially holds for episodes 3 and 4. These episodes contained predominantly convective precipitation. By making use of the effective $Z - R$ relations shown in Fig. 9 much larger precipitation values are obtained. Furthermore, since these two episodes contain the rain with the highest rainfall intensities observed for this event at the location of the radar, these larger estimates have a strong effect on the total accumulated rainfall. The Marshall-Palmer relationship overestimates the amount of rainfall only during three episodes. Episodes 6 and 7 produce much less rain than would be expected using the Marshall-Palmer relation (M-P), while episode 2 yields slightly less rain compared to M-P for higher reflectivities. These results clearly illustrate the impact of DSD variability on the effective $Z-R$ relations and the limited effectiveness of a applying a single static relation.





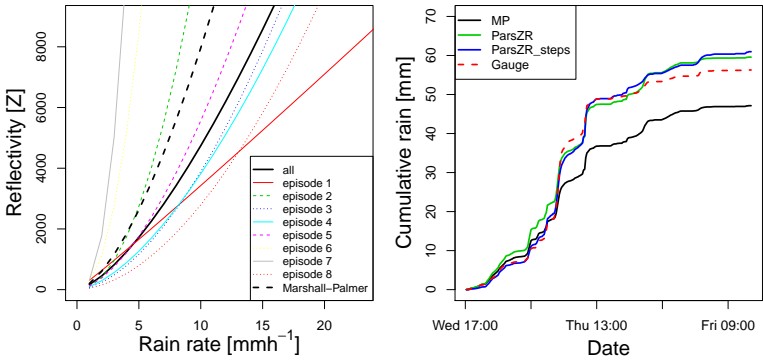

**Figure 10.** Left panel: All non-linear fits together with the Marshall-Palmer $Z - R$ relation. Right panel: Accumulated rain for different $Z - R$ relations applied to reflectivity data that are corrected for calibration, static clutter, and wet radome attenuation.

### 4.4.2 Application of $Z - R$ relations

The effects of applying the derived $Z - R$ relations are shown in the right panel of Fig. 10. In this figure three different approaches have been applied. First, the standard Marshall-Palmer relationship is used (similar to figure 7). As a second approach, the event based $Z-R$ relation obtained from all $Z$ and $R$ data collected by the Parsivel during this event, (top-left panel of Figure 9). Using a single representative $Z - R$ relationship leads to considerably more precipitation (59.6 mm) than M-P (47.1  mm) and results in an overestimation of about 6% in total accumulation as compared to the nearby rain gauge. As a third procedure, the eight individual, optimal intra-event power-law relationships were applied. This leads to a total precipitation accumulation of 61.0 mm, an overestimation by 8.3%. The results clearly reveal the positive impact of applying a event-based $Z-R$ relationship, which for the current event could successfully be derived from surface observations obtained by a single disdrometer. It should also be noted that even though considerable variations in the DSD are observed between the different episodes, leading to variations in the effective $Z-R$ relationship, this does not further improve the event-based precipitation estimates.

### 4.5 Verification of the correction methods

Several statistical parameters were selected to give a summary of the overall quality of the radar improvements compared to the rain gauge. These are the sum, relative mean bias and the coefficient of variation.

The sum is defined as the accumulated rain sum found from the different radar corrections:

$$\text{Sum} = \sum_{i=1}^{N} R(i)\Delta t \tag{5}$$





where $R(i)$ is the rain intensity of the radar for the current time step and $\Delta t$ is the time interval in hours (i.e., 0.1667 h for 10-minute values).

The relative mean bias is defined as the mean difference between 10-minute rainfall intensities from the radar and the rain gauge, normalized by the average gauge intensity:

$$\text{Bias} = \frac{\sum\limits_{i=1}^{N}(R(i)-G(i))}{\sum\limits_{i=1}^{N}G(i)} * 100\% \tag{6}$$

Where $G(i)$ is defined as the rain intensity of the gauge for the current time step.

The coefficient of variation (CV) is defined as the standard deviation of the gauge-radar differences of 10-minute intensities, normalized by the average 10-minute gauge intensities:

$$CV = \frac{\sqrt{\frac{1}{N}\sum\limits_{i=1}^{N}\left(R(i)-G(i)-\frac{1}{N}\sum\limits_{i=1}^{N}(R(i)-G(i))\right)^2}}{\frac{1}{N}\sum\limits_{i=1}^{N}G(i)} * 100\% \tag{7}$$

Table 2 illustrates the above mentioned statistics for several combinations of corrections. The operational product (applied M-P and Doppler corrected clutter filter) gives rise to a large underestimation by the radar and the worst performance. Not performing any kind of clutter correction, leads to better results as compared to the operational product, both in total rainfall, standard deviation as well as bias. If a wet-radome correction is applied to the static clutter corrected images all statistics improve and the difference between rain gauge and radar decreases from 38.8% to 16.3%. By converting the corrected reflectivity data using the Parsivel-inferred $Z - R$ relation further improvements in the quality of the radar product are obtained. Now the differences become very small, both in terms of bias and coefficient of variation. This is also apparent from the difference in rainfall accumulations. While the radar now slightly overestimates the gauge this value is much closer to the rainfall accumulation than when using M-P. Finally, the use of a different Parsivel-derived $Z - R$ relation for each episode gives a larger overestimation compared to the rain gauge. The coefficient of variation also decreases with each improvement. The effect would have been even larger when time steps without precipitation would be removed from the data. This reduces the CV for the full correction from 123% to 81%.

The final results for both correction methods based on the Parsivel-derived $Z - R$ relations are comparable to the observations by the rain gauge. The comparison between both instruments is remarkably close, especially in case one takes the differences between both instruments, their effective sampling volumes and the impact of elevation differences into account.

# 5  Summary and Conclusions

In the current study, close-range quantitative precipitation estimation (QPE) by radar was analyzed. By focusing specifically on regions close to the radar, the effect of rain-induced attenuation, VPR, and non-uniform beam filling are expected to be small,





**Table 2.** Rainfall sum [mm] from the radar together with relative mean bias [%] and coefficient of variation (CV) [%] of 10-minute rainfall intensities from rain gauge and the radar for different correction combinations. Here 'Operational' means the operational Doppler clutter corrected data, 'raw' means the uncorrected data, 'static' the static clutter corrected data and 'wet' the wet-radome corrected data. 'M-P' is the rain rate derived using the Marshall-Palmer equation. 'Pars' means rain rate derived using the $Z - R$ relation found from all data of the Parsivel. Finally 'Pars-steps' is the same, but for all episodes of the event the derived $Z - R$ values are used.

|  | Sum [mm] | Relative mean bias [%] | CV [%] |
|---|---|---|---|
| M-P Operational | 25.8 | -54.1 | 224 |
| M-P raw | 39.0 | -30.8 | 206 |
| M-P static | 34.5 | -38.8 | 205 |
| M-P wet+static | 47.1 | -16.3 | 157 |
| ParsZR wet+static | 59.6 | 5.9 | 139 |
| ParsZR-steps wet+static | 61.0 | 8.3 | 123 |

allowing to focus on errors due to calibration, clutter, and wet radome attenuation, as well as $Z - R$ variability, specifically. It was found that for this event the operational clutter-corrected radar product underestimated the rainfall accumulation by 54.1% compared to the rain gauge using a standard Marshall-Palmer $Z - R$ relation. The operational time-domain Doppler clutter filter used by KNMI is shown to erroneously filter some of the rain as well. By correcting radar volume data for clutter using a

simple static clutter filter the underestimation reduces to 38.8%. Further improvement is obtained when the data are corrected for wet radome attenuation by using a stable clutter target close to the radar as reference. These two corrections jointly with a correction for calibration error give an optimal estimated reflectivity from the radar. Applying M-P this resulted in a 16.3% underestimation with respect to the rain gauge.

Finally, the $Z - R$ relation was analyzed in detail to investigate if this could improve results. This was done by fitting a

power-law function using $Z$ and $R$ values obtained from the Parsivel disdrometer and applying this to the fully corrected radar reflectivities. This resulted in a slight overestimation of 5.9%. Additionally, the event was split up in 8 different episodes based on DSD-data and radar images. A dedicated $Z - R$ relation was derived for each episode, again based on the Parsivel data. The optimal $Z - R$ relation was found to be highly variable over the event. Applying these $Z - R$ relations to the fully corrected radar reflectivity data gave a slightly larger overestimation compared to the rain gauge. The standard deviation of the difference

between gauge and radar, using a different $Z - R$ relation for each episode, is slightly lower than when applying a single $Z - R$ relation for the entire event.

The results presented in this work clearly show the rainfall estimation capabilities of the radar have tremendous potential, if errors can be properly corrected for. Even at close ranges from the radar, multiple sources of error are shown to significantly affect radar rainfall estimates. The multiplicative nature of most of these errors means that their effect on rainfall estimates

is greatest at high rainfall intensities. It is shown here that using a time-domain Doppler clutter filter on all radar pixels causes significant underestimation. An operational algorithm that is more selective in clutter filtering (e.g. CMD, see Hubbert





et al., 2009), or using a Doppler filter with spectral reconstruction (e.g. Nguyen and Chandrasekar, 2013) will likely reduce this problem. Application of the technique used to correct for the wet-radome attenuation to an entire radar image is not recommended because wet-radome attenuation may be strongly dependent on azimuth, which probably depends on the wind speed and direction). It is relevant to study this in more detail because wet-radome attenuation can cause major (i.e. 3-4 dB; see Fig. 7) underestimation of precipitation.

The availability of a disdrometer enabled the derivation of intra-event $Z$-$R$ relations for selected periods. The current study showed that this lead to improved rainfall estimates as obtained from the radar, instead of using the standard Marshall-Palmer relationship. The current work only focussed on the impact of an event based $Z$−$R$ relationship close to the radar. Recently, Hazenberg et al. (2014, 2015) for summertime precipitation extended this approach to the whole radar domain, identifying

the benefits and limitations of using the disdrometer information. For convective precipitation the potential benefit of these instruments was small. This is because the locations of these cells do not correspond to the location of the disdrometer. However, for widespread stratiform precipitation, by making use of observations obtained by a single disdrometer, a much better correspondence with the rain gauges was observed as compared to applying a single static $Z$−$R$ relationship. These results show both the possibilities and limitations of making use of disdrometer observations to derived information on the

event-based $Z$-$R$ relationship.

*Acknowledgements.* Financial support for this work was provided by the Netherlands Space Office (NSO) and Netherlands Organization for Scientific Research (NWO) through grant EO-058. The authors would also like to thank Marijn de Haij of KNMI for providing the disdrometer data used in this paper.



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
