# Peer review of "Close-range radar rainfall estimation and error analysis"

_Atmospheric Measurement Techniques, 2016_

## Short Comment (SC1) · 23 Mar 2016

The calibration of the radar using the radar equation is only valid for range gates in the far-field of the antenna, i.e. beyond the Fraunhofer distance $d=2D^2/\lambda$, D=antenna diameter, lambda= wavelength. For the radar considered in this paper the Fraunhofer distance is 670m, i.e. the analyzed range gates are in the far-field. Another potential error originates from the T/R limiter of the radar. After the transmitter pulse is finished the limiter needs some time to recover. The older the limiter the longer is the recovery period. It is possible that there are a few tenth of a dB up to 1dB of attenuation even after several microseconds. Since the the distance of the first range gate is only 1km some additional attenuation is quite possible if the limiter is old. Unfortunately the attenuation of the limiter cannot be measured in the same way as the waveguide insertion losses.

---

## Author Comment (AC1) · 5 Apr 2016

We thank Frank Gekat for the valuable addition about other sources of error that may play a role. It is clear that the data that we have analyzed is from a range that is in the far field of the radar. The T/R limiter may indeed cause some tenths of a dB additional attenuation at the close range we're considering in this paper. We're confident that ageing of the T/R limiter has not degraded its performance, as KNMI operationally monitors this possible degradation (see Beekhuis and Leijnse, 2012). The limited effect of the T/R limiter is confirmed by our final results, where the final rainfall estimates are seen to correspond well with disdrometer and rain gauge measurements.

We will include a note in the revised paper about the fact that we're in the far field of the radar, and we'll briefly discuss the effect of the T/R limiter.

[Figure]

References:

Beekhuis, H. and H. Leijnse (2012), An operational radar monitoring tool. In: proceedings of the 7th European Conference on Radar in Meteorology and Hydrology, Toulouse, France, paper 47DQ, http://www.meteo.fr/cic/meetings/2012/ERAD/extended_abs/DQ_029_ext_abs.pdf.
* * *

---

## Referee Comment (RC1) · M. Montopoli (Referee) · 6 Apr 2016

Journal: AMT

Title: Close-range radar rainfall estimation and error analysis

Author(s): R. van de Beek et al.MS No.: amt-2016-77

The paper describes a case study in the Netherlands where weather radar and disdrometer acquisitions are compared each other in a configuration where vertical variability of DSD as well as path attenuation can be considered negligible.
The final goal is to put evidence (and quantify) on two aspects:
1. Quality in the procedures for the radar signal processing (ground clutter removal, wet radome loss compensation, absolute calibration od the reflectivity factor).
2. Temporal variability of DSD.

The paper reads very well and although the conclusions of the manuscript are not surprisingly new the presentation is good and arguments are convincing me. Using the right level of complexity the Authors quantify the impact of the various radar processing steps to better mimic the evolution of rain accumulations registered by the nearest rain gauge.

I recommend for publication after minor revision.

**Comments/suggestions**

- I am expecting a positive impact of an event based Z-R in absence of VPR effects. In the presence of VPR effects we have a problem of repressiveness of the Z-R relationships aloft with respect to those at the ground.
Do you have the chance to check at the temporal variation of the VPR at the considered site (i.e. using the rest of the radar antenna elevations) to produce errors, which would be representative of the non-optimal configuration (i.e. when observing rain precipitation at some distance above the ground)? In other word what happen considering Z at different elevations?

- pag. 1. line 4.  Abstract . : "5) vertical profile of reflectivity"  more in general I would say vertical variability  of DSD. Not only the reflectivity is affected  by the vertical variations, although in this paper only the reflectivity is ised.

 - pag 2, line 5 On the NUBF I would cite
ALEXANDER V. RYZHKOV,  The Impact of Beam Broadening on the Quality of Radar Polarimetric Data, JOURNAL OF ATMOSPHERIC AND OCEANIC TECHNOLOGY MAY 2007

After, at line 33 of the same page,  I would explain more which  are the effects of NUBF on Z (reduction?). Have you checked NUBF effects for the considered case of study. Is the spectral width available for the considered event? Please explain.

- pag 6, line 30. Reading this sentence it seems that you have not considered the effects of the calibration, ground clutter and wet radome as well. This is not the case of course. I think the phrase need to me modified.

- pag 8. It would be useful to show the map of the clutter map  cited in the text.

-pag 9. line 4. "Subtraction of the mean value of Z (i.e. not in dBZ)". I would expect a subtraction in dBZ, which implies a division in linear units. Am I wrong?

- pag. 12 figure 9. Could you please a different color for the black curve?

- pag 12. eq. 2. How is calculated sigma_B in your equation? Please explain in the main texts.

---

## Referee Comment (RC2) · N.I. Fox (Referee) · 6 Apr 2016

General Comments

This paper does an excellent job of separating the potential errors involved in estimating rainfall rates from single polarization radar reflectivity observations. The concentration on a single range gate (very) close to the radar is an original way of removing the effects caused by elevated beam heights such as brightband and wind drift. I believe it adds significantly to the methodology of operational rainfall retrieval using radar.

I believe that the paper is worthy of publication but could benefit from a more complete discussion of a couple of issues.

Specific comments

[Figure]

I think the authors should discuss to some extent the uncertainties in the calculations of Z and R from the observed DSD. Most notably for a 1 minute sample how many drops are typically observed? An example of this is shown to some extent in figure 4 (in a very nice way), but as we know that Z is very sensitive to the concentration of larger drops if there are small numbers of such drops counted in 1 minute then the uncertainty in Z can be large. Also the authors use only the Parsivel data in their analysis but the drop size bins of the Parsivel are pretty wide for large drops, so assigning a suitable size to an individual (or small number) of drops in these bins is problematic. Can the authors comment on this?

One of the final conclusions is that using the overall Z-R found from all the events using Z and R values calculated from the observed DSDs improves rainfall retrieval. How is this different from finding a local Z-R from comparisons of radar reflectivity and rain gauges? On the other hand, I can see the advantage of the Z-R steps method of intra-event DSD calibration, but can't think how this could be applied operationally, and even if it were it would have limited areal applicability as the inherent assumption is that the rain DSD is spatially variable. Can the authors please comment on these issues?

I'd like to see the information shown in the legends of the plots in figure 9 presented in a table. In particular a table of a and b coefficients of the Z-R would make the range of value combinations easier to assess (the print is also very small and hard to see). I think this is important information as it would allow readers to see how close these come to the alternatives to the Marshall-Palmer Z-R, such as the US National Weather Service convective Z-R.

Also, for the relationships found from these plots, is there statistical evidence (R2 values for example) that the nonlinear fit is better than the linear. I would agree that the nonlinear fits look better and often the statistical tests are inconclusive due to the concentration of points near the origin, but it would help justify the choice.

Page 9: Is it not possible to use a second elevation to fill in clutter-contaminated range

gates? Would this be better than trying to subtract what could be a varying clutter signal from the observed reflectivity?

Technical Corrections

Page 8, line 1: Is this calibration exactly 1 dB or was this value approximate or chosen for simplicity? Page 8, line 8: 'speed" should be "velocity".

Page 9, line 8: Should read "an operational Doppler. . ."

---

## Editor Comment (EC1) · G. Vulpiani (Editor) · 8 Apr 2016

**Comments on "Close-range radar rainfall estimation and error analysis" by R. van de Beek et al.**

**Error sources**

Some error sources (the main, as stated in the manuscript) affecting the radar rainfall estimate are listed in the Abstract and Introduction. In my opinion, there are some additional sources that should be mentioned: **1) beam blockage, 2) W-LAN interferences and 3) hail contamination**.

- The first is definetly among the main error sources in complex terrain scenarios, even more than attenuation (at least at C-band) considering the latter as a transient phenomenon whose effects over medium to long cumulation intervals might not be detrimental, even if not corrected. Beam blockage, despite quantifiable and correctable to a given extent, affects the adopted scan strategy, the height of measurement above ground, the impact of melting hydrometeors. This is the main problem caused by orography, ground clutter returns being relatively easy to identify.
- The second is being a really bothersome issue, it is not rare to observe returns higher than 30 dBZ.
- Hail, that is definitely not a rare hydrometeor type, especially for some climatological conditions, heavily affects radar observations in terms of both scattering and absorption enhancement. At C-band, small hail is responsible for resonance, including attenuation enhancement in case of melting hail that can lead to signal extintion even in short range path (see the related works by Meteo France and NSSL among others).

The list of error sources should unequivocally mention the vertical variability of precipitation (in terms of size, particle size distribution and refractive index) in place of VPR, the latter being a direct effect of the first. This is correctly done in page 2 lines 27-32 but not in the abstract.

**Z-R derivation**

This is the most "critical" comment I have.
For the sake of clarity, I think the authors should mention and possibly deal with some of the issues related to the use of the Parsivel observations. Most of them have been clearly outlined by Thurai et al. (2011), Tokay et al. (2013).

Is there any impact on the present analysis?

Also, to my knowledge, the Parsivel processing assumes a constant raindrop axis ratio for diameters larger than 5 mm. This assumption might have an impact on the convective rain events you have considered. Could you please say something on this topic?

**Miscellaneous**

- Referring to **attenuation correction** methodologies the Authors did not mention very important works for ground-based radar applications, i.e., Bringi et al., (1990), Testud et al., (2000) (derived by the rain profiling algorithms), that represent the basic platforms for any existing operational attenuation correction algorithms. If I am not wrong, the approach proposed by Delrieu et al., uses the returns by fixed obstacles (mountains) to constraint the correction. However, it can only be applied to a limited portion of the radar domain.

- Figure 9, currently the title of vertical axes is "Reflectivity (Z)", maybe the units of Z should be explicited.

**References**

Bringi, V. N., V. Chandrasekar, N. Balakrishnan, and D. S. Zrnic, 1990: An examination of propagation effects in rainfall on radar measurements at microwave frequencies. J. Atmos. Oceanic Technol.,7,829–840.

Testud, J., E. Le Bouar, E. Obligis, and M. Ali-Mehenni, 2000: The rain profiling algorithm applied to polarimetric weather radar. J. Atmos. Oceanic Technol.,17,332–356

Thurai, M., Petersen, W. A., Tokay, A., Schultz, C., and Gatlin, P.: Drop size distribution comparisons between Parsivel and 2-D video disdrometers, Adv. Geosci., 30, 3-9, doi:10.5194/adgeo-30-3-2011, 2011.

Tokay, A., W. A. Petersen, P. Gatlin, M. Wingo, 2013: Comparison of Raindrop Size Distribution Measurements by Collocated Disdrometers. *Journal of Atmospheric and Oceanic Technology*, **30**, 1672–1690, doi: 10.1175/JTECH-D-12-00163.1.

---

## Referee Comment (RC3) · Anonymous Referee #3 · 13 Apr 2016

This manuscript is an excellent overview of the most important error sources affecting radar-based QPE, with a focus on observations collected at short distances from the radar. This overview is well presented and the reader is guided step-by-step through the case study of August 2010. The manuscript reads well, it is well suited for AMT and it should be published after very minor corrections.

General comment: It would be interesting to have the confirmation that similar outcomes can be expected for other rainfall events, and thus that the results are general (for the given set-up). The manuscript should still be based on the current precipitation event (as it reads very well this way), but could anything be added in the appendix?

Short comments: 1) Clearly the manuscript focuses on non-polarimetric radars. Could the author define in the text (briefly) the concept of polarimetric radar?

2) Page 2, Lines 20-25. Here it may be interesting to talk about some polarimetric methods for attenuation correction.

3) Page 2, Line 32. It could be stated that above the melting layer precipitation is usually under-estimated given the dielectric properties of ice (snow).

4) Page 4, Line 5. For expert readers, it may be useful to mention the "generation" of Parsivel used.

5) Page 5, Line 1. Could you give an order of magnitude of how similar those measurements are? Are they another order of magnitude with respect to the corrections proposed later on?

6) Page 8, Line 1. How often is operationally re-calibrated the radar, by means of this procedure?

7) Page 8: Clutter correction. Could you give the filter width of the notch filter operationally implemented?

8) Figures 6 and 7: i would prefer a lot to see the 3 panels vertically aligned, with the same width.

9) Page 9, Line 3. If not dBZ, please give the units here (units are given only later on in section 4.4.2)

10) Page 11, Z - R relations. Among the different corrections, this one seems more difficult to implement real-time. It is not the main focus of the manuscript, but could you spend some words about the potential rea-time implementation of those corrections?

11) Page 12, Lines 5-10. Could you define also D as the equivalent volume diameter?

---

## Referee Comment (RC4) · M. Rico-Ramirez (Referee) · 27 Apr 2016

Manuscript: "Close-range radar rainfall estimation and error analysis" Authors: van de Beek et al. doi:10.5194/amt-2016-77

The paper attempts to quantify some of the error sources in weather radar observations (such as ground clutter, radome attenuation and Z-R variability) by comparing radar observations at very short range (1-2 km) with raingauge and disdrometer measurements. The paper is very interesting and AMT readers would benefit from this paper. The paper is well written and it should be published after the authors address some minor issues as discussed below. 1- An important source of "error" between radar and raingauge measurements is due to the fact that radar observations are areal (in fact volume) rainfall measurements whereas raingauges provide point rainfall mea-

surements (Kitchen and Blackall, 1992; Ciach & Krajewski, 1999; Bringi et al, 2011). This produces some differences when comparing both sets of observations even at short range because the radar spatial resolution is relatively large (1km along-range in your case) compared to the raingauge sampling area. Please comment on this and give an indication of how much of the observed difference between radar and gauge measurements is due to the point to area variance. 2 - Another source of error in radar measurements is due to the fact that the radar provides instantaneous measurements whereas raingauges provide measurements integrated in time. Operational weather radars usually perform volume scans (i.e. several PPI scans at different elevations) and therefore the sampling time interval of the surface radar rainfall measurements is relatively large (5min in your case). Errors due to the sampling time interval can be large especially in convective situations (see e.g. Fabry et al, 1994). Interpolation techniques can help to mitigate this (e.g. using nowcasting). How much of the observed difference between radar and gauge measurements is due to the radar temporal sampling? Please comment on this. 3 – Radome attenuation. The correction for the radome attenuation was performed using a fixed clutter target, but ignoring the effect of wetting of the clutter target and precipitation at the clutter location. The authors also highlighted the fact that radome attenuation depends on wind speed and direction. Please give an indication on how reliable is the proposed radome attenuation correction, perhaps by making reference to other papers. 4 – Z-R variability. The study concludes that applying an event-based Z–R relationship obtained from disdrometer observations improves the radar rainfall estimation. Although this is true for the location under consideration, it is well known that the Z-R equation changes in space and time. Please comment on this. 5 – The study was performed on a limited data set (only 3 days), but it is likely that the radar errors will depend on the precipitation type (e.g. stratiform rain, convective rain, winter storms, etc). Please comment on this.

Minor Comments:

Fig 3. Please be consistent with the use of colours in fig 3 (radar measurements

[Figure]

were shown in red in top panel and in black in bottom panel). Similarly for gauge measurements.

Page 8 "zero-isoDop"?

Section 4.1. It is unclear which azimuthal angle is used for the comparisons.

References.

Ciach, G. J., and W. F. Krajewski, (1999): On the estimation of radar rainfall error variance. Adv. Water Resour., 22, 585–595.

Fabry, F et al (1994): "High resolution rainfall measurements by radar for very small basins: the sampling problem reexamined," Journal of Hydrology , vol. 161, pp. 415–428.

Bringi, VN, et al (2011): 'Rainfall Estimation with an Operational Polarimetric C-band Radar in the UK: Comparison with a Gauge Network and Error Analysis' Journal of Hydrometeorology, vol 12, pp. 935 – 954.

Kitchen, M., and R. M. Blackall, 1992: Representativeness errors in comparisons between radar and gage measurements of rainfall. J. Hydrol., 134, 13–33.

---

## Author Comment (AC2) · 22 Jun 2016

**Reply to comments of Mario Montopoli on Close-range radar rainfall estimation and error analysis**

*The paper describes a case study in the Netherlands where weather radar and disdrometer acquisitions are compared each other in a configuration where vertical variability of DSD as well as path attenuation can be considered negligible. The final goal is to put evidence (and quantify) on two aspects:*
*1. Quality in the procedures for the radar signal processing (ground clutter removal, wet radome loss compensation, absolute calibration od the reflectivity factor).*
*2. Temporal variability of DSD.*
*The paper reads very well and although the conclusions of the manuscript are not surprisingly new the presentation is good and arguments are convincing me. Using the right level of complexity the Authors quantify the impact of the various radar processing steps to better mimic the evolution of rain accumulations registered by the nearest rain gauge.*
*I recommend for publication after minor revision.*

The authors would like to thank Mario Montopoli for his comments. Below we will give a reaction to each separate comment.

1. *I am expecting a positive impact of an event based Z-R in absence of VPR effects. In the presence of VPR effects we have a problem of repressiveness of the Z-R relationships aloft with respect to those at the ground.*
   *Do you have the chance to check at the temporal variation of the VPR at the considered site (i.e. using the rest of the radar antenna elevations) to produce errors, which would be representative of the non-optimal configuration (i.e. when observing rain precipitation at some distance above the ground)? In other word what happen considering Z at different elevations?*

   For the event that is studied in this paper, the effects of VPR can be considered insignificant for the lowest 1500 m. This can be seen in the Fig. 1 for the Contoured Frequency by Altitude Diagrams (CFADs). For the total of all phases, but also for the individual phases this is mostly the case. The drop distribution and the division in phases shown in Fig. 1 are shown in Fig. 2. Note that for this study the effect of VPR is negligible because we're only using data at very close range, where the center of the beam is less than 100 m high.

[Figure]

Figure 1. Accumulated Contoured Frequency by Altitude Diagrams. Upper left is the average for all episodes and the following panels represent individual episodes within the event.

[Figure]

**Figure 2. Illustration of the drop size distribution measured by the distrometer. Upper panel shows the number of raindrops per diameter size and the lower panel shows the median and 10-90 percentile drop diameter.**

2. *pag. 1. line 4. Abstract . : "5) vertical profile of reflectivity" more in general I would say vertical variability of DSD. Not only the reflectivity is affected by the vertical variations, although in this paper only the reflectivity is used.*

   The authors agree that this would reflect a more general statement and this will be changed.

3. *pag 2, line 5 On the NUBF I would cite ALEXANDER V. RYZHKOV, The Impact of Beam Broadening on the Quality of Radar Polarimetric Data, JOURNAL OF ATMOSPHERIC AND OCEANIC TECHNOLOGY MAY 2007*

   This reference will be added

4. *After, at line 33 of the same page, I would explain more which are the effects of NUBF on Z (reduction?). Have you checked NUBF effects for the considered case of study. Is the spectral width available for the considered event? Please explain.*

   The effect of non-uniform beam filling of course depends on the vertical profile of the DSD. In general, if we're interested in the rain intensity (or reflectivity) at ground level, the effect could either be an enhancement of Z in case of a bright band, or a reduction of Z in case of shallow precipitation or measurements above the melting layer (where Z decreases with height). We assume that the effects of NUBF are negligible in this study because of the close range we're using (the 1-degree beam is only about 50 m wide at a 3-km range).

5. *pag 6, line 30. Reading this sentence it seems that you have not considered the effects of the calibration, ground clutter and wet radome as well. This is not the case of course. I think the phrase need to me modified.*

   We will rephrase this sentence to better reflect that we're not taking these error sources into account because of the close range we're using.

6. *pag 8. It would be useful to show the map of the clutter map cited in the text.*

The principle of a clutter map is usually applied to each individual pixel of a radar image, but in this study only a single bin was used. We did not derive a clutter map for the entire radar image. The term "clutter map" might be a bit misleading because of this and we will reword to avoid confusion. The baseline clutter level for this location is shown in Fig. 6 of the manuscript.

7. *pag 9. line 4. "Subtraction of the mean value of Z (i.e. not in dBZ)". I would expect a subtraction in dBZ, which implies a division in linear units. Am I wrong?*

We deliberately choose to subtract Z in linear units. The rationale behind this is that we assume that the power returned by a clutter target and that returned by precipitation are independent, and can be added. Since the retuned power and Z (in linear units) are linearly related, we substract the derived clutter power in linear units.

8. *pag. 12 figure 9. Could you please a different color for the black curve?*

We will use a different color for these lines.

9. *pag 12. eq. 2. How is calculated sigma_B in your equation? Please explain in the main texts.*

We assume here that the backscattering cross-section is:

$$\sigma_B = \frac{\pi^5 |K|^2}{10^6 \lambda^4} D^6$$

So that the equation for Z will be:

$$Z = \int_0^\infty D^6 N(D) dD$$

We will modify Eq. (2) accordingly, so that the backscattering cross-section will no longer appear.

---

## Author Comment (AC3) · 22 Jun 2016

**Reply to comments of Neil Fox on Close-range radar rainfall estimation and error analysis**

*This paper does an excellent job of separating the potential errors involved in estimating rainfall rates from single polarization radar reflectivity observations. The concentration on a single range gate (very) close to the radar is an original way of removing the effects caused by elevated beam heights such as bright band and wind drift. I believe it adds significantly to the methodology of operational rainfall retrieval using radar.*
*I believe that the paper is worthy of publication but could benefit from a more complete discussion of a couple of issues.*

The authors would like to thank Neil Fox for his comments. Below we will give a reaction to each separate comment.

*1. I think the authors should discuss to some extent the uncertainties in the calculations of Z and R from the observed DSD. Most notably for a 1 minute sample how many drops are typically observed? An example of this is shown to some extent in figure 4 (in a very nice way), but as we know that Z is very sensitive to the concentration of larger drops if there are small numbers of such drops counted in 1 minute then the uncertainty in Z can be large. Also the authors use only the Parsivel data in their analysis but the drop size bins of the Parsivel are pretty wide for large drops, so assigning a suitable size to an individual (or small number) of drops in these bins is problematic. Can the authors comment on this?*

DSD sampling effects (both in terms of number of drops per unit time and diameter class widths) are indeed relevant aspects. We will add a short discussion to the paper regarding these issues. Even though this issue is relevant, we do not think it will greatly influence the outcome of this paper (see e.g. Salles and Creutin, 2003, Tokay et al., 2005, Uijlenhoet et al., 2006, and Leijnse and Uijlenhoet, 2010)

Salles, C. and Creutin, J.-D.: Instrumental uncertainties in Z−R relationships and raindrop fall velocities, J. Appl. Meteorol., 42, 279–290, 2003.
Tokay, A., Bashor, P. G., and Wolff, K. R.: Error characteristics of rainfall measurements by collocated Joss-Waldvogel disdrometers, J. Atmos. Ocean. Technol., 22, 513–527, 2005.
Uijlenhoet, R., Porrà, J. M., Sempere Torres, D., and Creutin, J.-D.: Analytical solutions to sampling effects in drop size distribution measurements during stationary rainfall: Estimation of bulk rainfall variables, J. Hydrol., 328, 65–82, 2006.
Leijnse, H. and Uijlenhoet, R.: The effect of reported high-velocity small raindrops on inferred drop size distributions and derived power laws, Atmos. Chem. Phys., 10, 6807-6818, doi:10.5194/acp-10-6807-2010, 2010.

*2. One of the final conclusions is that using the overall Z-R found from all the events using Z and R values calculated from the observed DSDs improves rainfall retrieval. How is this different from finding a local Z-R from comparisons of radar reflectivity and rain gauges? On the other hand, I can see the advantage of the Z-R steps method of intra-event DSD calibration, but can't think how this could be applied operationally, and even if it were it would have limited areal applicability as the inherent assumption is that the rain DSD is spatially variable. Can the authors please comment on these issues?*

Our main reason for using DSD data to infer optimal Z-R relations is that these would be independent of all other potential radar error sources. This allows us to really investigate the effect of all of the different error sources. Using a radar-gauge comparison to infer a Z-R relation would result in a Z-R relation that compensates for all other remaining error sources.

Regarding the operational applicability of using separate Z-R relations for different types of events, we agree that this is not possible without an unrealistically dense network of disdrometers. Our purpose was mainly to quantify the magnitude of the errors when using improper Z-R relations. Having said this, one could of course think of an application where data from a limited network of disdrometers

could be used to derive event type-specific Z-R relations (for e.g. convective, stratiform, etc. rain) that could then be operationally applied (reverting back to climatological relations if no rain occurs over any of the disdrometers).

*3. I'd like to see the information shown in the legends of the plots in figure 9 presented in a table. In particular a table of a and b coefficients of the Z-R would make the range of value combinations easier to assess (the print is also very small and hard to see). I think this is important information as it would allow readers to see how close these come to the alternatives to the Marshall-Palmer Z-R, such as the US National Weather Service convective Z-R.*

We will include a table with Z-R coefficients and exponents. Note that the left-hand panel of Fig. 10 also shows a summary of these relations.

*Also, for the relationships found from these plots, is there statistical evidence (R2 values for example) that the nonlinear fit is better than the linear. I would agree that the nonlinear fits look better and often the statistical tests are inconclusive due to the concentration of points near the origin, but it would help justify the choice.*

We have no statistical evidence that the nonlinear fit is better than the linear fit. The problem with such statistical evidence is that it is probably easy to find statistics that would show that either linear or logarithmic fitting would be best. The main reason for using non-linear fits is that in most practical applications high rainfall intensities are much more important than low rainfall intensities. Using logarithmic fitting would result in the difference between 0.01 mm/h and 0.1 mm/h being as important as the difference between 10 mm/h and 100 mm/h. Furthermore, due to the highly skewed distribution of rainfall intensities, low intensities naturally receive much weight. We do not want to increase this weight even further by using logarithmic fitting.

*Page 9: Is it not possible to use a second elevation to fill in clutter-contaminated range gates? Would this be better than trying to subtract what could be a varying clutter signal from the observed reflectivity?*

Using a higher (uncontaminated) elevation to fill in a clutter-contaminated pixel would indeed be helpful. However, for this event and pixel, we found that most of the lower elevations are still contaminated by some (side lobe) clutter, so we did not use this technique.

*Page 8, line 1: Is this calibration exactly 1 dB or was this value approximate or chosen for simplicity?*

The value was indeed not exactly 1 dB, but the uncertainty in this number is such (a few tenths of a dB) that rounding to 1 dB is justified.

*Page 8, line 8: 'speed" should be "velocity".*

We will modify this.

*Page 9, line 8: Should read "an operational Doppler*

We will modify this.

---

## Author Comment (AC4) · 22 Jun 2016

**Reply to comments of Anomynous Referee #3 on Close-range radar rainfall estimation and error analysis**

*This manuscript is an excellent overview of the most important error sources affecting radar-based QPE, with a focus on observations collected at short distances from the radar. This overview is well presented and the reader is guided step-by-step through the case study of August 2010. The manuscript reads well, it is well suited for AMT and it should be published after very minor corrections.*

The authors would like to thank the referee for these comments. Below we will give a reaction to each separate comment.

*General comment: It would be interesting to have the confirmation that similar outcomes can be expected for other rainfall events, and thus that the results are general (for the given set-up). The manuscript should still be based on the current precipitation event (as it reads very well this way), but could anything be added in the appendix?*

Testing other events is unfortunately beyond the scope of this study. Because of the different types of precipitation that were observed in this event, we think that it is reasonable to assume that the results presented here hold for more than just this event, and that they are quite general, We will make a remark about this in the paper.

*Short comments:*
*1) Clearly the manuscript focuses on non-polarimetric radars. Could the author define in the text (briefly) the concept of polarimetric radar?*

We will add a very brief description of polarimetric radar.

2) Page 2, Lines 20-25. Here it may be interesting to talk about some polarimetric methods for attenuation correction

*We will mention polarimetric attenuation correction in the paper.*

*3) Page 2, Line 32. It could be stated that above the melting layer precipitation is usually under-estimated given the dielectric properties of ice (snow).*

For this study we stay well below the melting layer, but we will change the text to reflect both the uncertainties within and above the melting layer.

*4) Page 4, Line 5. For expert readers, it may be useful to mention the "generation" of Parsivel used.*

The Parsivel data we used was from a 1$^{st}$ generation Parsivel. We will add this information to the paper.

*5) Page 5, Line 1. Could you give an order of magnitude of how similar those measurements are? Are they another order of magnitude with respect to the corrections proposed later on?*

The differences between the two disdrometers in terms of rainfall accumulations can be seen in the bottom panel of Fig. 3. For the largest part of the event, the rainfall intensities from both disdrometers are approximately equal. For the most intense part of the event, the LPM seems to record more than the Parsivel and the rain gauge. For clarity and conciseness, we will remove all references to the LPM from the paper, as it does not add much.

*6) Page 8, Line 1. How often is operationally re-calibrated the radar, by means of this procedure?*

The radar receiver calibration is monitored daily using the sun. We only recalibrate the radar if we see very large deviations or discontinuities. In practice, we have never needed to recalibrate the radar beside that which is carried out as part of annual maintenance. We do, however, spot radar system

part malfunctioning through this method of monitoring, which helps us to keep a stable radar. With respect to the transmitter calibration, we monitor the transmitted power as well as the returns of stable clutter targets on a daily basis. Again, we do not operationally adjust the transmitter calibration based on this outside of the annual calibration, but we do find system faults

*7) Page 8: Clutter correction. Could you give the filter width of the notch filter operationally implemented?*

The width of the employed Doppler notch filter is approximately 1 m/s (we use dual-PRF, and have a Doppler notch width of 0.84 m/s for the lower PRF and 1.13 m/s for the higher PRF).

*8) Figures 6 and 7: i would prefer a lot to see the 3 panels vertically aligned, with the same width.*

The upper left panels are time series for a much longer period than the other two. The reason why the upper-left panels of these figures have a longer time span is that we want to show what happens in dry weather preceding and following the event. The reason for plotting the intensity graph across the entire width of the figure is that it contains much more detail than the accumulation graph. So we will keep these figures as they are.

*9) Page 9, Line 3. If not dBZ, please give the units here (units are given only later on in section 4.4.2)*

We will add the units of Z [$mm^6$ $m^{-3}$].

*10) Page 11, Z - R relations. Among the different corrections, this one seems more difficult to implement real-time. It is not the main focus of the manuscript, but could you spend some words about the potential rea-time implementation of those corrections?*

Our purpose was mainly to quantify the magnitude of the errors when using non-optimal Z-R relations. Having said this, one could of course think of an application where data from a limited network of disdrometers could be used to derive event type-specific Z-R relations (for e.g. convective, stratiform, etc. rain) that could then be operationally applied (reverting back to climatological relations if no rain occurs over any of the disdrometers). See also our reply to the comment made by Neil Fox about this.

11) Page 12, Lines 5-10. Could you define also D as the equivalent volume diameter?

Yes, D could be defined as the equivalent spherical volume diameter.

---

## Author Comment (AC5) · 14 Jul 2016

**Reply to comments of Gianfranco Vulpiani (editor) on Close-range radar rainfall estimation and error analysis**

The authors would like to thank Gianfranco Vulpiani for providing useful comments that will improve our paper. Below we respond to each comment.

**Error sources**

*Some error sources (the main, as stated in the manuscript) affecting the radar rainfall estimate are listed in the Abstract and Introduction. In my opinion, there are some additional sources that should be mentioned: **1) beam blockage, 2) W-LAN interferences and 3) hail contamination**.*

- *The first is definitely among the main error sources in complex terrain scenarios, even more than attenuation (at least at C-band) considering the latter as a transient phenomenon whose effects over medium to long accumulation intervals might not be detrimental, even if not corrected. Beam blockage, despite quantifiable and correctable to a given extent, affects the adopted scan strategy, the height of measurement above ground, the impact of melting hydrometeors. This is the main problem caused by orography, ground clutter returns being relatively easy to identify.*

- *The second is being a really bothersome issue, it is not rare to observe returns higher than 30 dBZ.*

- *Hail, that is definitely not a rare hydrometeor type, especially for some climatological conditions, heavily affects radar observations in terms of both scattering and absorption enhancement. At C-band, small hail is responsible for resonance, including attenuation enhancement in case of melting hail that can lead to signal extinction even in short range path (see the related works by Meteo France and NSSL among others).*
  *The list of error sources should unequivocally mention the vertical variability of precipitation (in terms of size, particle size distribution and refractive index) in place of VPR, the latter being a direct effect of the first. This is correctly done in page 2 lines 27-32 but not in the abstract.*

We agree that these sources of error are also important. We will discuss them in both the abstract and the introduction.

**Z-R derivation**
*This is the most "critical" comment I have.*
*For the sake of clarity, I think the authors should mention and possibly deal with some of the issues related to the use of the Parsivel observations. Most of them have been clearly outlined by Thurai et al. (2011), Tokay et al. (2013).*
*Is there any impact on the present analysis?*
*Also, to my knowledge, the Parsivel processing assumes a constant raindrop axis ratio for diameters larger than 5 mm. This assumption might have an impact on the convective rain events you have considered. Could you please say something on this topic?*

We are indeed aware of the problems with first-generation Parsivel disdrometers as reported by Tokay et al., (2013). The most notable Parsivel overestimates occur for higher rainfall intensities, and large raindrops. Tokay et al. (2013) note that the Parsivel starts to overestimate the number of drops larger than 2.44 mm diameter when intensities exceed 2.5 mm/h and the drop concentrations exceed 400 per minute. The number of drops in this diameter class (i.e., larger than 2.44 mm) is limited throughout the event (see Fig. 4 of the manuscript), so we expect the effect to be minor. The rainfall peak of episode 3 is most prone to this

error, but even then the number of drops is limited, especially compared to the number of small drops. So even for this peak, the effect will be limited. We will add a short description of the problem and the fact that the effects are very minor to the text (including references to the papers mentioned above).

**Miscellaneous**

- *Referring to **attenuation correction** methodologies the Authors did not mention very important works for ground-based radar applications, i.e., Bringi et al., (1990), Testud et al., (2000) (derived by the rain profiling algorithms), that represent the basic platforms for any existing operational attenuation correction algorithms. If I am not wrong, the approach proposed by Delrieu et al., uses the returns by fixed obstacles (mountains) to constraint the correction. However, it can only be applied to a limited portion of the radar domain.*

We will include these important references in the paper.

- *Figure 9, currently the title of vertical axes is "Reflectivity (Z)", maybe the units of Z should be explicited.*

We will modify the figure so that the units of Z are displayed: "$[mm^6 \, m^{-3}]$".

**References**

*Bringi, V. N., V. Chandrasekar, N. Balakrishnan, and D. S. Zrnic, 1990: An examination of propagation effects in rainfall on radar measurements at microwave frequencies. J. Atmos. Oceanic Technol.,7,829–840.*

*Testud, J., E. Le Bouar, E. Obligis, and M. Ali-Mehenni, 2000: The rain profiling algorithm applied to polarimetric weather radar. J. Atmos. Oceanic Technol.,17,332–356*

*Thurai, M., Petersen, W. A., Tokay, A., Schultz, C., and Gatlin, P.: Drop size distribution comparisons between Parsivel and 2-D video disdrometers, Adv. Geosci., 30, 3-9, doi:10.5194/adgeo-30-3-2011, 2011.*

*Tokay, A., W. A. Petersen, P. Gatlin, M. Wingo, 2013: Comparison of Raindrop Size Distribution Measurements by Collocated Disdrometers. Journal of Atmospheric and Oceanic Technology, **30**, 1672–1690, doi: 10.1175/JTECH-D-12-00163.1.*

---

## Author Comment (AC6) · 14 Jul 2016

**Reply to comments of Miguel Rico-Ramirez on Close-range radar rainfall estimation and error analysis**

*The paper attempts to quantify some of the error sources in weather radar observations (such as ground clutter, radome attenuation and Z-R variability) by comparing radar observations at very short range (1-2 km) with raingauge and disdrometer measurements. The paper is very interesting and AMT readers would benefit from this paper. The paper is well written and it should be published after the authors address some minor issues as discussed below.*

The authors would like to thank Miguel Rico Ramirez for the positive comments, and the valuable suggestions for improving the paper.

*1- An important source of "error" between radar and raingauge measurements is due to the fact that radar observations are areal (in fact volume) rainfall measurements whereas raingauges provide point rainfall measurements (Kitchen and Blackall, 1992; Ciach & Krajewski, 1999; Bringi et al, 2011). This produces some differences when comparing both sets of observations even at short range because the radar spatial resolution is relatively large (1km along-range in your case) compared to the raingauge sampling area. Please comment on this and give an indication of how much of the observed difference between radar and gauge measurements is due to the point to area variance.*

The large differences in sampling volumes is indeed a cause for differences between radar and rain gauge measurements of rain. The fact that we're using data close to the radar limits the size of the radar measurement volume (in azimuth and elevation directions anyway), but it is still orders of magnitude larger than a rain gauge orifice. However, we think that the differences between the two is compensated by the fact that rain gauges integrate in time (see comment 2), whereas radars provide instantaneous measurements. If we assume that Taylor's hypothesis of frozen turbulence holds for rain over a 5-minute period (similar to the assumptions behind the interpolation techniques to correct for advection) the effective scale of a rain gauge, translated to an instantaneous measurement, is much closer to that of a radar pixel (depending on the advection speed of the event, see Fabry et al., 1994). There are of course still differences, as we state on line 24 of p.15. We believe that a more thorough discussion of these differences is outside the scope of this paper.

*2 - Another source of error in radar measurements is due to the fact that the radar provides instantaneous measurements whereas raingauges provide measurements integrated in time. Operational weather radars usually perform volume scans (i.e. several PPI scans at different elevations) and therefore the sampling time interval of the surface radar rainfall measurements is relatively large (5min in your case). Errors due to the sampling time interval can be large especially in convective situations (see e.g. Fabry et al, 1994). Interpolation techniques can help to mitigate this (e.g. using nowcasting). How much of the observed difference between radar and gauge measurements is due to the radar temporal sampling? Please comment on this.*

See also our reply to comment 1. We fully agree with this comment that for advection speeds greater than 1 km per 5 minutes (i.e., 3.3 m/s) the temporal resolution of the radar is too coarse to capture the temporal variability of rain. For this event, the advection speed is indeed greater than 3.3 m/s. However, the space-time structure of the precipitation for this event was such that this had only a minor effect on our results. We will state this in the Discussions section, along with a reference to Fabry et al. (1994).

*3 – Radome attenuation. The correction for the radome attenuation was performed using a fixed clutter target, but ignoring the effect of wetting of the clutter target and precipitation at the clutter location. The authors also highlighted the fact that radome attenuation depends on wind speed and direction. Please give an indication on how reliable is the proposed radome attenuation correction, perhaps by making reference to other papers.*

The quality of our wet radome correction method is indeed uncertain due to the effects mentioned by the reviewer. The most uncertain part is the effect of the wetting of the clutter target. We don't know how this affects the reflectivity. Rain in the same pixel as the clutter target is not likely to influence results because the clutter target generates reflectivities (60 dBZ) that far exceed those generated by rain in this event. Because the method we use is not intended to be used in an operational setting (as stated on lines 2-5 on p.17) we feel that a further discussion of the quality of this correction method is not needed for the purpose of this paper.

*4 – Z-R variability. The study concludes that applying an event-based Z–R relationship obtained from disdrometer observations improves the radar rainfall estimation. Although this is true for the location under consideration, it is well known that the Z-R equation changes in space and time. Please comment on this.*

We have already included a discussion on this on lines 6-15 on p.17. With respect to the spatial variability of the drop see distribution, the applicability of disdrometers is limited when the inter-disdrometer distance is greater than the scale on which the DSD varies. However, the temporal variability of the DSD at the location of the disdrometer is well-captured on time scales of 5 minutes.

*5 – The study was performed on a limited data set (only 3 days), but it is likely that the radar errors will depend on the precipitation type (e.g. stratiform rain, convective rain, winter storms, etc). Please comment on this.*

We agree with the fact that errors can be different for different precipitation types. The nice thing about this event is that it contained both convective and stratiform rain. Of course, there are more types of events that could occur (we had no hail or other solid precipitation on the ground during this event). We will note this in the conclusions section of the paper.

*Minor Comments:*
*Fig 3. Please be consistent with the use of colours in fig 3 (radar measurements were shown in red in top panel and in black in bottom panel). Similarly for gauge measurements.*

We will change the colours used in the top panel of Figure 3 so that they are consistent with the bottom panel and other figures in the paper.

*Page 8 "zero-isoDop"?*

We've explained "zero-isoDop" on lines 8-9 of p.8, right after we first mention this term.

*Section 4.1. It is unclear which azimuthal angle is used for the comparisons.*

In Section 4.1 we do not use a given azimuth, but we use all measured sun interferences over that day to obtain a robust estimate of the receiver calibration and possible offsets in antenna pointing angles. More details on this method can be found in Holleman et al. (2010).

*References.*
*Ciach, G. J., and W. F. Krajewski, (1999): On the estimation of radar rainfall error variance. Adv. Water Resour., 22, 585–595.*
*Fabry, F et al (1994): "High resolution rainfall measurements by radar for very small basins: the sampling problem reexamined," Journal of Hydrology , vol. 161, pp. 415– 428.*
*Bringi, VN, et al (2011): 'Rainfall Estimation with an Operational Polarimetric C-band Radar in the UK: Comparison with a Gauge Network and Error Analysis' Journal of Hydrometeorology, vol 12, pp. 935 – 954.*
*Kitchen, M., and R. M. Blackall, 1992: Representativeness errors in comparisons be- tween radar and gage measurements of rainfall. J. Hydrol., 134, 13–33.*